# Xanthohumol—A Miracle Molecule with Biological Activities: A Review of Biodegradable Polymeric Carriers and Naturally Derived Compounds for Its Delivery

**DOI:** 10.3390/ijms25063398

**Published:** 2024-03-17

**Authors:** Ewa Oledzka

**Affiliations:** 1Chair and Department of the Pharmaceutical Chemistry and Biomaterials, Faculty of Pharmacy, Medical University of Warsaw, 1 Banacha Str., 02-097 Warsaw, Poland; eoledzka@wum.edu.pl; Tel.: +48-22-572-07-55; 2Yappco Sp. z o.o., Ignacego Mościckiego 1 Str, 24-110 Puławy, Poland

**Keywords:** xanthohumol, *Humulus lupulus*, chalcone, biodegradable polymers, drug carriers, targeted therapy, naturally derived carrier

## Abstract

Xanthohumol (Xn), a prenylated chalcone found in Hop (*Humulus lupulus* L.), has been shown to have potent anti-aging, diabetes, inflammation, microbial infection, and cancer properties. Unfortunately, this molecule has undesirable characteristics such as inadequate intake, low aqueous solubility, and a short half-life. To address these drawbacks, researchers have made numerous attempts to improve its absorption, solubility, and bioavailability. Polymeric drug delivery systems (PDDSs) have experienced significant development over the last two decades. Polymeric drug delivery is defined as a formulation or device that allows the introduction of a therapeutic substance into the body. Biodegradable and bioreducible polymers are the ideal choice for a variety of new DDSs. Xn formulations based on biodegradable polymers and naturally derived compounds could solve some of the major drawbacks of Xn-based drug delivery. In this regard, the primary concern of this study is on presenting innovative formulations for Xn delivery, such as nanoparticles (NPs), nanomicelles, nanoliposomes, solid lipid nanoparticles (SLNs), and others, as well as the received in vitro and in vivo data. Furthermore, this work describes the chemistry and broad biological activity of Xn, which is particularly useful in modern drug technology as well as the cosmetics industry. It is also important to point out that the safety of using Xn, and its biotransformation, pharmacokinetics, and clinical applications, have been thoroughly explained in this review.

## 1. Introduction

Phytochemicals present in plants, herbs, and spices have recently shown promise not just for their antibacterial and antiviral activities, but also for the treatment of cancer and chemoprevention [1]. Hops are gaining importance in this context. Hops (*Humulus lupulus* L., family *Cannabaceae*), or more accurately, the female plant’s flowers (cones), are widely used in the brewing industry to provide a characteristic flavour and bitterness to beer. Depicting the traditional use of hop, it has been used for the preservation and flavouring of alcohol-based beverages since 200 A.D., with therapeutic treatments beginning as early as the ninth century [2]. When it was discovered in 79 A.D., it was popular as a vegetable, but it was later utilised as a dye and a culinary flavour ingredient [3,4]. It was also used in the production of coarse textiles and paper. Hop plants were traditionally used as a sedative for treating sleeplessness and restlessness, as well as for relieving ear discomfort, toothache, and loss of appetite [2]. Currently, hop products are often used as dietary supplements because of their hypnotic and anxiolytic characteristics, and they are utilized in the treatment of postmenopausal symptoms in women. Furthermore, they are broadly applied in the cosmetic and pharmaceutical industries, owing to their antibacterial and antiviral properties [5,6]. Hop versatility lies in the fact that it possesses a wide range of physiologically active compounds [7,8]. Terpenes, bitter acids, and chalcones are the three main structural types of chemical substances identified in hop mature cones. Hops are also rich in flavonol glycosides (kaempferol, quercetin, quercitrin, rutin) and catechins (catechin gallate, epicatechin gallate) [3,9]. Hundreds of terpenoid components were found in the volatile oil (0.3–1.0% of hop strobile weight): principally caryophyllene, farnesene, and humulene (sesquiterpenes) and myrcene (monoterpene) [10]. Bitter acids (5–20% of hop strobile weight) are phloroglucinol derivatives categorised as α-acids and β-acids. Hops contain bitter acids in a complex combination with varying compositions and concentrations [3]. Aside from the volatile oil and bitter acids, many prenylflavonoids have been identified in hop cones [9]. The most significant chemical is chalcone xanthohumol (Xn, Figure 1) (up to 1% in dried hop cones), which can be converted to prenylfavanone isoxanthohumol (IX) at a higher pH value and under thermal treatment [9]. As a result, IX is the primary prenylflavonoid found in beer. Other chalcones isomerize to corresponding flavanones at concentrations 10–100-times lower than Xn. A chalcone known as xanthogalenol (XG) has only been found in a few hop varieties. Desmethylxanthohumol (DMX), also known as 2′,4′,6′,4-tetrahydroxy-3′-C-prenylchalcone, has been determined to be the precursor of most flavonoids found in hops. Through chemical isomerization, the major estrogen of hops is produced, identified as the 1:1 racemate (±)-8-prenylnaringenin (8-PN), along with racemic 6-prenylnaringenin (6-PN). In humans, 8-PN has been shown to be derived from IX via activation by the gut microbiota or liver cytochrome P450 enzymes. Thus, estrogen-inactive Xn possesses the estrogenic potential of being converted to IX and then to 8-PN [3,11,12,13].

Although the structure of Xn was discovered in 1957, its favourable pharmacological properties were not recognised until the 1990s [14]. Numerous research carried out in the recent years have demonstrated its anti-inflammatory, antibacterial, antiviral, antifungal, antiplasmodial, and anticancer properties [15,16,17]. Taking into account that Xn possesses multiple characteristics in addition to its natural and relatively non-toxic nature, it could be a very promising active substance for further in vivo and clinical investigations as well as possible applications in a range of therapies.

However, it is noteworthy to point out that delivery of Xn to the target site is a challenge due to its high hydrophobicity, low stability, high photosensitivity, short half-life, and poor oral bioavailability [6]. Therefore, it is necessary to create innovative formulations that overcome all of the restrictions associated with Xn oral bioavailability and transport to the target site without losing at the biotransformation sites. Innovative Drug Delivery Systems (DDSs) have opened the way for biological molecules to be delivered to the target site by eliminating the major barriers to their transportation.

The research and development of biodegradable polymers represent a 50-year-long medical revolution, resulting in significant scientific advances in drug delivery, tissue engineering, biomaterials, and others while bringing together chemists, engineers, biologists, and physics collectively in a novel and cooperative way [18]. The short half-life of many drugs, along with non-specific distribution and toxicity, has been a primary driving force in the development of Polymer Drug Delivery Systems (PDDSs). The successful clinical translation of the first macro- and micro-drug delivery systems has resulted in the development of controlled-release nanodrug delivery platforms that may overcome pharmacological limitations and offer considerable benefits over traditional dosage forms [19,20]. Synthetic methods, fabrication procedures, and mathematical models for the study of the controlled drug release mechanisms have opened up an opportunity of developing defined polymer nanoparticles (NPs) DDSs able to provide localized and sustained delivery, enabling an improvement in the drug therapeutic index. The controlled release of drugs and the high adaptability of PDDSs provide multiple benefits [21,22]. The introduction of controlled-release polymeric NPs has driven novel investigations into synthetic approaches, bioconjugation techniques, and an increasing number of scientific papers on the usage of chemical processes that result in modifications to the structure, size, and degradation rates of NPs. The ability to fine-tune the physicochemical characteristics of polymeric NPs and include targeting components in their design has allowed new generations of controlled-release polymeric NPs to navigate a complex and chemically rich environment in vivo [18,23,24]. With more knowledge of biological processes in disease states, the design of NPs controlled-release PDDSs has advanced beyond conventional release mechanisms to apply local biochemical changes in abnormal disease states to trigger and activate drug release. To optimise the efficacy of biodegradable polymers in drug delivery, there is growing interest in combining biologically responsive elements into the overall polymer design to achieve more biologically controlled therapeutic results. Biodegradable and bioabsorbable polymers are a great option for a variety of innovative DDSs. The delivery component of the systems has been created using bioabsorbable polymers such as poly(lactic acid) (PLA) and poly(glycolic acid) (PGA), as well as their copolymers. Whether DDSs use a biodegradable implant to deliver drugs subcutaneously or within the body, biodegradable and bioabsorbable polymers provide a safe framework for delivering drugs without causing adverse effects on the body [25,26].

In light of the foregoing, in this review, I conducted an extensive literature search to find data on biodegradable polymers and naturally derived compounds as carriers for Xn delivery, as well as the aforementioned keywords, using the databases Scopus, Web of Science, PubMed, and Google Scholar. Aside from highlighting the chemistry and structure of Xn, as well as its broad biological activity, which is not only particularly useful in modern drug technology but also in the cosmetics industry, I intended to investigate the safety of Xn use, as well its biotransformation, pharmacokinetics, and clinical applications. In the final part of the paper, I collected and discussed a variety of Xn formulations based on biodegradable polymers and other natural and nontoxic compounds, as well as in vitro and in vivo data with additional investigations conducted by researchers.

## 2. Chemical and Biological Properties of Xn

### 2.1. Structure and Chemistry of Xn

Xn (1-(2,4-dihydroxy-6-methoxy-3-[3-methylbut-2-en-1-yl] phenyl)-3-(4 hydroxyphenyl) prop-2-en-1-one) is usually generated as solid yellow crystals with a melting point of 172 °C (Pubchem CID 639665) [6]. It is insoluble in water and in petroleum ether, and it can be crystallized in 50% alcohol, 50% acetone, acetic acid, chloroform, benzene and toluene. It dissolves in strong alkali and in sulphuric acid, and it shows no optical activity. The mean value of its elementary analyses is 72.0% C and 6.18% H (calculated for C_2l_H_22_O_5_: 71.2% C and 6.21% H) [14].

Xn’s structure comprises of flavonoids chain with aromatic rings A and B arranged in the *trans*-position and substituted with hydroxyl and methoxy groups, one unsaturated double bond α, β, and a prenyl unit (Figure 1). Xn has biological activity due to the existence of α,β-unsaturated ketone group. This compound’s lipophilicity is increased by substituting the A ring with a prenyl unit and the -OCH_3_ group, resulting in a high affinity for membranes in biological systems [4,6,27].

The spectroscopic and spectrophotometric characteristics of this compound are as follows:

^1^H NMR (DMSO-*d*_6_): 1.61 (3H, s, H-5″), 1.70 (3H, s, H-4″), 3.13 (2H, d, H-1″), 3.87 (3H, s, C6′O-CH_3_), 5.14 (1H, m, H-2″), 6.08 (1H, s, H-5′), 6.84 (1H, m, H-3 and H-5), 7.58 (1H, m, H-2 and H-6), 7.67 (1H, d, *J* = 15.6 Hz, H-β), 7.77 (1H, d, H-α), and 14.69 (C2′-OH).

^13^C NMR (DMSO-*d*_6_): 17.7 (C-4″), 21.1 (C-1″), 25.5 (C-5″), 55.8 (C6′O-CH_3_), 91.0 (C-5′), 104.6 (C-1′), 107.4 (C-3′), 116.0 (C-3 and C-5), 123.1 (C-2″), 123.8 (C-α), 126.1 (C-1), 130.0 (C-3″), 130.5 (C-2 and C-6), 142.6 (C-β), 160.0 (C-4), 160.6 (C-6′), 162.4 (C-4′), 164.7 (C-2′), and 191.7 (C=O). UV (MeOH): λ_max_ = 368.1 nm [28].

FTIR (cm^−1^): 974 trans -CH=CH-, 1027 substituted benzene, 1145 ν(C-C) aromatic ring, ν(-OH), 1228 δν(-OH), 1292 ν(C-C) aromatic ring, 1346 δ(-OH), 1470 δ(C-C), 1545 trans CH=CH-, 1606 νC=O, 2854 ν_s_(-OCH_3_), 2967 ν_as_ (-OCH_3_), 3189 ν(-OH), ν(C-H) in aromatic ring [29].

It is also important to highlight the most recent literature investigation, in which scientists examined into the storage stability and degradation mechanism of Xn [30]. The authors discovered that this molecule degraded rapidly when exposed to high temperatures and light. Furthermore, at high temperatures, Xn was susceptible to isomerization, hydration, and ortho-cyclization processes, resulting in the creation of a variety of degradation products. According to the data, Xn degraded according to a first-order kinetic model.

### 2.2. Biological Properties of Xn

Xn has a broad variety of pharmacological activities against diabetes, inflammation, viral infection, cardiovascular disease, and cancer [16]. Xn and its related flavone, IX, reduce adipogenesis by blocking preadipocyte differentiation, decreasing lipogenic proteins, and enabling mature adipocytes to undergo mitochondrial apoptosis [31,32]. Furthermore, Xn treatment raised the AMP-activated protein kinase (AMPK) signalling pathway, inhibiting lipogenesis in a type 2 diabetes mellitus (T2DM) mouse model. Its consumption reduced body weight gain and enhanced plasma lipid profile, with substantial improvements in insulin resistance and glucose tolerance. By modulating glucose and lipid pathways, an Xn-enriched diet may be able to alleviate diabetic-associated metabolic abnormalities [33]. The modulating action of Xn on the farnesoid X receptor (FXR) in vitro and in vivo has also been examined by Nozawa et al. [34]. The authors proved that in the transient transfection test, Xn boosted the activity of the human bile salt export pump (BSEP) promoter-driven luciferase in a dose-dependent manner. Xn-fed KK-A^y^ mice had decreased levels of plasma glucose, plasma, and liver triglyceride. They also had lower water consumption, lower white adipose tissue weights, and higher plasma adiponectin levels, showing that Xn prevented diabetes in KK-A^y^ mice. Xn also addresses skin aging and pigmentation while promoting photoprotection, thus making it an effective component for the cosmetics industry. In the light of this, the effects of Xn on melanogenesis in MNT-1 human melanoma cells and normal human melanocytes from darkly pigmented skin (HEM-DP) were studied in the publication [35]. Melanosome degradation was also investigated in human keratinocytes (HaCaT). The researchers discovered that Xn reduced the production of melanin in MNT-1 cells while increasing intracellular tyrosinase activity without changing Reactive Oxygen Species (ROS) levels. Xn inhibited cellular tyrosinase activity in HEM-DP cells without affecting synthesis of melanin. Xn also limited melanosome export by reducing dendrite number and length. Further tests in HaCaT cells demonstrated that Xn caused melanosome degradation without cytotoxicity [35]. The authors of the other study [36] explored Xn as an anti-aging substance by favourably regulating the extracellular matrix. They examined how Xn influenced the activities of elastase and matrix metalloproteinases (MMPs, MMPs 1, 2, and 9), as well as the expression of collagen, elastin, and fibrillins in dermal fibroblasts. It was discovered that Xn inhibited elastase and MMP-9 activities at low concentrations while activating MMP-1 and MMP-2 at higher concentrations. The research results were similar to those of ascorbic acid [36]. It is also worth noting the work of Kang and colleagues, who investigated the effects of Xn and related compounds on the production of interleukin (IL)-12, the most important factor driving T helper 1 immune response [37]. Xn had the greatest inhibitory effect on IL-12 production in macrophages stimulated by lipopolysaccharide (LPS) or LPS/interferon-γ. Furthermore, it was determined whether Xn reduced skin inflammation. Chronic allergic contact dermatitis, an experimental model for psoriasis, was used to assess the anti-inflammatory effects of Xn in vivo. It was established that Xn treatment reduced the degree of ear thickening caused by oxazolone [37]. Furthermore, five strains, *Propionibacterium acnes*, *Staphylococcus epidermidis*, *Staphylococcus aureus*, *Kocuria rhizophila*, and *Staphylococcus pyogenes*, were chosen for testing the biological activities of Xn extract on acne vulgaris. Xn demonstrated strong inhibitory activity against all strains. Furthermore, it showed moderate to strong anticollagenase inhibitory activity. Antioxidant capacity was also assessed using seven different methods based on various ROS. Xn had the highest activity in both total oxygen radical absorbance capacity and singlet oxygen absorbance capacity [38].

The modern cosmetics industry provides its customers with a wide range of products, many of which are holistic in nature. This is achieved by combining synergistic active ingredients in novel ways or by utilising the multifunctionality of a single component. Xn should be considered a substance that meets both of the above criteria because, in addition to the anti-aging and anti-pigmentation properties mentioned above, as well as photoprotection, it also possesses antibacterial, antiviral, antimalarial, and antifungal properties.

#### 2.2.1. Antibacterial Activity

Previously, the ability of Xn to inhibit the growth of Gram-positive *Staphylococcus aureus*, a pathogen commonly found in pneumonia and sepsis, was compared to antibiotic activity against *Escherichia coli*. This compound was discovered to inhibit *Escherichia coli* proliferation while being a potent inhibitor of *Staphylococcus aureus* with a minimal inhibitory concentration (MIC) of 17.7 μM [39]. In addition, Xn was tested against three strains of *Streptococcus* and its activity was compared to that of some essential oils commonly found in anticaries mouth washes [40]. Xn demonstrated antimicrobial activity against *Streptococcus mutans*, *Streptococcus salivarius*, and *Streptococcus sanguis* in a disc diffusion assay. Xn (140 nmol) produced similar zones of inhibition against all three strains at a dose of 50 μg per disc as thymol (333 nmol). Furthermore, the MIC of Xn from hop extracts ranged from 10 to 50 μg/mL for *Clostridium perfringens* strains and from 15 to 60 μg/mL for *Bacteroides fragilis* strains, indicating that this molecule is more effective against these two strains than α- and β-acids [40]. Based on Xn’s potent antibacterial effects on *Clostridium difficile*, as well as its low toxicity and favourable pharmacokinetics, this molecule was being considered as a promising therapeutic agent for the treatment and prevention of *Clostridium difficile* infections. The data showed that the MIC and minimum bactericidal concentrations of Xn ranged from 32 to 107 μg/mL and from 40 to 107 μg/mL, respectively [40,41].

Latest in vivo studies on the bioavailability of Xn in rats revealed that this chalcone and its metabolites are primarily excreted in faeces [42]. As a result, it was interesting to evaluate if Xn could influence the intestinal microbiota. Hanske et al. used polymerase chain reaction denaturing gradient gel electrophoresis (PCR-DGGE) to investigate the composition of rat intestinal microbiota. In comparison to untreated controls, daily Xn applications to male and female Sprague Dawley (SD) rats for 4 weeks had no effect on the diversity of the faecal microbial community [43]. Added to that, the purified extracts, which comprise α- and β-acids, as well as Xn, were examined in vivo for their inhibitory action against *Clostridioides difficile*, a prominent pathogen responsible for nosocomial gastrointestinal infections in humans [44,45]. The researchers developed a rat model for this pathogen attacking the peroral intestinal tract. Both Xn and hop acids were discovered to have substantial antibacterial activity. The Xn application, in particular, showed antibacterial activity as well as a reduction in local inflammatory symptoms in the large intestine [44].

#### 2.2.2. Antifungal Activity

There have been only a few investigations on the antifungal activity of Xn. However, it was discovered that this compound was a potent antifungal agent against five human pathogenic fungi, namely *Trichophyton mentagrophytes*, *Trichophyton rubrum*, *Candida albicans*, *Fusarium oxysporum*, and *Mucor rouxianus* [39]. Xn inhibited the growth of the dermatophytic fungi *Trichophyton mentagrophytes* and *Trichophyton rubrum* more effectively than the positive control griseofulvin (MIC: 6.25 μg/mL). Xn also inhibited *Mucor rouxianus* (MIC = 50 μg/mL). *Candida albicans* and the problematic human pathogen *Fusarium oxysporum*, on the other hand, were unresponsive to Xn (MIC > 200 μg/mL).

#### 2.2.3. Antiviral Activity

In 2004, Buckwold and coworkers tested Xn against a number of DNA and RNA viruses in vitro [46]. As RNA viruses, bovine viral diarrhoea virus (BVDV), a surrogate model for hepatitis C virus, and human rhinovirus (HRV) were included. In addition, the DNA herpesviruses cytomegalovirus (CMV) and herpes simplex virus types 1 and 2 (HSV-1 and -2) were used to test antiviral activity of Xn. The inhibitory effects of this chalcone against BVDV (NADL strain in MDBK cells), HRV (rhinovirus 14 strain in MRC-5 cells), HSV-1 (F strain in Vero cells), and HSV-2 (MS strain in Vero cells) were assessed using cell-based assays designed to assess inhibition of cytopathic effects (CPE). In MRC-5 cells, CMV (strain AD169) was tested in a plaque reduction assay. The authors discovered that Xn potently inhibited the growth of BVDV, CMV, HSV-1, and HSV-2. The half-maximal inhibitory concentrations (IC_50_) values of Xn to inhibit viral replication ranged from 1.5 to 2.7 μg/mL. Xn, on the other hand, exhibited no antiviral activity against HRV. An Xn-enriched extract was also tested in the same study. The Xn in the extract appeared to account for almost all of the extract’s antiviral activity, as the therapeutic indices TI (TC_50_/IC_50_) of this compound against BVDV, HSV-1, and HSV-2 were comparable to those of the Xn-enriched extract [46]. Additionally, Wang and coworkers investigated the ability of Xn to inhibit various steps required for HIV-1 replication [47]. In C8166 lymphocytes infected with HIV-1_IIIB_, Xn was found to inhibit HIV-1-induced CPE, as well as viral p24 antigen production and reverse transcriptase activity as measures of active retroviral replication, with IC_50_ values of 2.3, 3.6, and 1.4 μM, respectively. With an IC_50_ value of 58.5 μM, this chalcone also inhibited HIV-1 replication in peripheral blood mononuclear cells. In this study, Xn had no effect on recombinant HIV-1 reverse transcriptase activity or HIV-1 entry into cells [47]. It is also worth noting that Liu et al. examined the therapeutic effect of Xn against highly pathogenic porcine reproductive and respiratory syndrome viruses (PRRSV) [48]. The authors discovered that Xn had a low IC_50_ value for inhibiting PRRSV infection in porcine primary alveolar macrophages (PAMs). Furthermore, it reduced the expression of interleukin (IL)-1, IL-6, IL-8, and tumour necrosis factor α in PRRSV-infected or lipopolysaccharide-treated PAMs. Xn effectively alleviated clinical signs, lung pathology, and inflammatory responses in pig lung tissues induced by highly pathogenic PRRSV infection, according to animal challenge experiments [48]

The pandemic of coronavirus disease 2019 (COVID-19) caused by severe acute respiratory syndrome coronavirus 2 (SARS-CoV-2) has resulted in substantial worldwide morbidity and mortality, with significant financial and social repercussions [49]. Coronavirus’s main protease (M^pro^) is crucial for viral replication and transcription, making it an appealing drug target for antiviral drug development. Lin et al. identified Xn to be a potent pan-inhibitor for various coronaviruses by targeting M^pro^, including β-coronavirus SARS-CoV-2 (IC_50_ value of 1.53 μM) and α-coronavirus PEDV (IC_50_ value of 7.51 μM). In enzymatic assays, Xn inhibited M^pro^ activities, while pretreatment with this molecule inhibited the SARS-CoV-2 and PEDV replication in Vero-E6 cells [49]. Furthermore, Dabrowski et al. investigated the impact of Xn on the inflammatory response and clinical outcome of COVID-19 patients [50]. As a result, adult patients with acute respiratory failure (PaO_2_/FiO_2_ less than 150) were examined. In this study, patients were randomly assigned into two groups: Xn—patients who received adjuvant treatment with Xn at a daily dose of 4.5 mg/kg body weight for 7 days, and C—controls. Observations were carried out at four points: immediately after admission to the intensive care unit (ICU), as well as on the third, fifth, and seventh day of treatment. The inflammatory response was measured using plasma IL-6 levels, neutrophil-to-lymphocyte ratios (NLR), platelet-to-lymphocyte ratios (PLR), C-reactive protein (CRP), and D-dimer levels. The death rate was calculated 28 days after ICU admission. The researchers discovered that 72 patients were able to participate in the study, and 50 were included in the final evaluation. The Xn group had a lower mortality rate and a shorter clinical course than the control group (20% vs. 48%, *p* < 0.05, and 9 ± 3 days vs. 22 ± 8 days, *p* < 0.001). Furthermore, Xn treatment significantly reduced plasma IL-6 concentrations (*p* < 0.01), D-dimer levels (*p* < 0.05), and NLR (*p* < 0.01) compared to standard treatment [50].

#### 2.2.4. Antimalarial Activity

Malaria, caused by protozoan parasites of the genus *Plasmodium*, constitutes one of the most common infectious diseases and an important problem for public health. Only four *Plasmodium parasite* species have the ability to infect humans. *Plasmodium falciparum* and *Plasmodium vivax*, however, cause the most severe forms of the disease [51]. Xn is one of the structural classes for which antiplasmodial/antimalarial activity has been reported with great interest in the scientific community in the recent times [39]. Herath et al. have previously reported the microbial transformation of Xn with *Cunninghamella echinulata* NRRL 3655 and the evaluation of these products for cytotoxicity towards mammalian cell lines as well as possible antimicrobial and antimalarial properties. The authors demonstrated that Xn and its microbial transformation products were antimalarial against *Plasmodium falciparium* D6 (chloroquine sensitive) and W2 (chloroquine resistant) strains. This chalcone was active against D6 and W2 strains, with IC_50_ values of 3.3 and 4.1 μg/mL, respectively [52]. In addition, the in vitro antiplasmodial activity of Xn and seven natural or semi-synthetic derivatives against two different strains of *Plasmodium falciparium* (chloroquine-sensitive strain poW and the multiresistant clone Dd2 using a [^3^H]hypoxanthine-incorporation assay), as well as their interaction with GSH-dependent haemin degradation, were evaluated. Xn was the most active chalcone in this study, with IC_50_ values of 8.2 ± 0.3 μm (poW) and 24.0 ± 0.8 μm (Dd2).

#### 2.2.5. Antiplatelet Activity

I would also like to briefly discuss Xn’s antiplatelet activity. It has previously been claimed that Xn is responsible for a decreased risk of cardiovascular disease by reducing the probability of platelet hyperreactivity during thrombosis [53]. As an instance, this chalcone considerably reduced ADP-induced blood platelet aggregation and fibrinogen receptor expression on platelet surfaces in C57BL/6J wild-type male mice [54]. Xn reduced platelet activation in C57/BL6 mice and SD rats by decreasing ROS accumulation and inhibiting the cellular damage factor mtDNA-induced DC-SIGN-dependent pathway, hence avoiding arterial and venous thrombosis without inducing bleeding [55]. Lee et al. found that Xn inhibited the phosphorylation of phospholipase C (PLC)γ2, p38 mitogen-activated protein kinase, extracellular antiregulated kinase 1/2, JNK1, and Akt, leading to decreased thromboxane A2 formation and Ca^2+^ mobilisation [41,56]. Nonetheless, Xin et al. discovered that Xn has antiplatelet and antithrombotic properties, which may reduce ROS accumulation by upregulating sirtuin1 (SIRT1) expression, followed by inhibition of mitochondrial dysfunction and a reduction in respiratory disorders as well mitochondrial hyperpolarization [55]. Another recent study found that this molecule could be beneficial in arrhythmias [57]. The effects of Xn (5–1000 nM) on Ca^2+^ signalling pathways were investigated in isolated rat ventricular myocytes incubated with Fluo-4AM using the perforated patch-clamp technique. The authors discovered that 5–50 nM Xn decreased the frequency of spontaneously occurring Ca^2+^ sparks and Ca^2+^ waves in control myocytes and cells subjected to Ca^2+^ overload. It also reduced the Ca^2+^ content of the Sarcoplasmic Reticulum (SR) and its rate of recirculation. Lastly, multiple investigations evaluating the potential of Xn in cancer treatment have discovered that it is effective in vitro and in vivo against a variety of cancer models: breast and cervical cancers, cholangiocarcinoma, glioblastoma, colon, colorectal, haematological, laryngeal, liver, ovarian, pancreatic, prostate, thyroid, and oesophageal cancers, melanoma, and oral squamous cell carcinoma have all been studied [1,4,6,15,17,57,58,59,60,61,62]. As a result, these findings will not be replicated in this review.

#### 2.2.6. Safety of the Xn Use

Throughout various activity studies, the impact of Xn on normal cells was additionally examined. In normal cells, such as human lung fibroblast cells (MRC-5), primary human hepatocytes, oligodendroglia-derived cells (OLN-93), and human skin fibroblasts, Xn demonstrated very low or no toxicity [17]. These outcomes indicated that Xn was specifically targeting cancer cells; however, Xn may be a safe and effective agent. Similar results were obtained in vivo. Vanhoecke and coworkers presented the findings of a four-week safety study of Xn in mice. The daily administration of 23 mg/kg body weight (b.w.) showed no signs of toxicity in bone marrow, liver, exocrine pancreas, kidneys, muscles, thyroid, or ovaries. The data indicate that oral administration of Xn to laboratory mice does not affect major organ functions and opens the gate for further safety studies in humans [63]. Furthermore, Hussong et al. explored the sub-chronic toxicity of Xn in female SD rats at daily doses up to 1000 mg/kg j.w. for 4 weeks, which causes mild hepatotoxicity in animals, but has no effect on reproduction or the development of two generations of offspring when given at a daily dose of 100 mg/kg b.w. Furthermore, the authors found that the treatment of male rats prior to mating significantly (*p* = 0.027) increased the sex ratio of male to female offspring. Overall, lifelong treatment at a daily dose of 100 mg/kg b.w. in a two-generation study did not affect the development of SD rats [64].

#### 2.2.7. Biotransformation, Pharmacokinetics and Clinical Applications of Xn

The available literature confirms the use of Xn, but there are several barriers between basic research and clinical practice [17]. The primary concern is its low bioavailability. As I mentioned above, Nookandeh and coworkers reported this occurrence during research on female SD rats, in which orally administered Xn was excreted within 48 h via faeces and urine [42]. Furthermore, Avula et al. investigated the resorption and metabolism of Xn in rat plasma, urine, and faeces following its oral or intravenous (i.v.) administration [65]. The authors found that plasma levels of Xn fell rapidly within 60 min after i.v. administration; no Xn was detected in plasma after its oral use. Moreover, Xn and its metabolites were excreted mainly in faeces within 24 h of administration. Meanwhile, Pang et al. determined that facilitated transport was not responsible for Xn uptake; rather, the accumulation in human colorectal adenocarcinoma (Caco-2) cells was presumably caused by specific binding to cytosolic proteins. This is mostly owing to its biotransformation in the gastrointestinal tract by hepatic enzymes and its accumulation in 70% on the apical side of Caco-2 cells. By binding to the cytosolic proteins, about 93% of intracellular Xn was localised in the cytosol. As a result, this agent failed to induce an effective therapeutic response at the intended site [66]. It is also worth noting that research was also conducted to determine the fundamental aspects of Xn absorption, distribution, and metabolism in male jugular vein-cannulated SD rats. As a result, the authors carried out a single-dose pharmacokinetics (PK) study at three oral dose levels and one i.v. dose level to assess bioavailability and dose-dependence of PK parameters. It was discovered that the dose-dependent bioavailability of Xn emphasises the importance of additional investigation into Xn metabolism in order to elucidate and optimise the potential health benefits of Xn and its metabolites [67]. Therefore, two years later, the same authors published the data collected in healthy men and women to determine basic PK parameters for Xn in order to establish dose–concentration relationships and predict dose–effect relationships in humans diagnosed with metabolic syndrome. According to the authors of this study, the Xn PK exhibited a distinct biphasic absorption pattern, with Xn and IX conjugates being the primary circulating metabolites following oral Xn administration in humans [68]. The next study attempted to determine whether, in healthy, normal-weight women, consuming a low dose of Xn, based on concentrations found in 250 mL beer consumed with a light breakfast, influences the LPS-dependent immune response of peripheral blood mononuclear cells (PBMCs) isolated following the intake of the hop compound [69]. It was proposed that acute consumption of low doses of Xn extract may suppress the LPS-dependent immune response of PBMCs in healthy women, and the beneficial effects of the hop compound were related to an inhibition of LPS binding to the cluster of differentiation 14 (CD14). At the end of this chapter, I would like to mention two more articles presenting the protocols for phase I and II triple-masked, placebo-controlled clinical trials on Xn microbiome and signature in healthy adults (the XmaS trial). The major goal of the first of them was to evaluate the clinical safety and tolerability of Xn in healthy and adult patients. The researchers also examined biomarkers reflecting inflammation, gut permeability, bile acid metabolism, and products of this compound metabolism in vivo, as well as the influence of Xn on the composition of gut microbial [70]. The performed preclinical studies have indicated that Xn has multiple therapeutic properties, involving modulating inflammatory pathways through the FXR agonists, inhibiting the nuclear factor-kappa B (NFκB) activation, regulating gut permeability, bile acid metabolism, and activating the nuclear factor erythroid 2 -related factor 2 (NRF2) to regulate antioxidant protein expression. The study demonstrated that Xn might impact the expression of numerous downstream genes in vivo, making it a possible Crohn’s disease (CD) treatment. Furthermore, it was discovered that Xn served as a prebiotic for intestinal microbiota, altering the gut microbiota and its bacterial metabolites [70]. Assuming in the second study, the same authors monitored the safety and tolerability of the same amount of Xn (24 mg daily) in adults with clinically active CD in a placebo-controlled phase II clinical trial, as well as the effect of Xn on inflammatory biomarkers, platelet function, CD clinical activity, and stool microbial composition [71]. The investigators have noticed some limitations since the trial began, but the findings of this study will add to the evidence base for the use of Xn for this disease patient population. This was owing, among other things, to the demographic distribution of Portland, which limited the generalizability of trial results; the fact that the population in the Portland area consumed a high volume of several types of compounds prohibited in this study, as well as the rarity of CD in the general population. However, this phase II trial yielded a considerable amount of data for a straight comparison with an otherwise healthy group in the phase I trial [71].

## 3. Biodegradable Polymers and Naturally Derived Compounds as the Carriers for Xn Delivery

When considering biodegradable polymers (natural and synthetic) for the development of PDDSs, it is critical to understand their various types and properties, as well as how these properties influence drug delivery and its release. A biodegradable polymer is one that degrades into simpler compounds when exposed to sunlight, moisture, and microorganisms in their natural environment [26]. Enzymes can break down polymer chains into smaller fragments, which can then be metabolised by microorganisms. Biodegradable polymers degrade by breaking down their chains into smaller molecules, such as monomers or oligomers, via chemical or biological processes. Several factors can influence their degradation process, including temperature, pH, moisture, and the presence of enzymes or microorganisms [18].

In general, several classes of polymers, including poly(esters), poly(ortho-esters), polyanhydrides, polycarbonates, poly(amides), poly(ester-amides), poly(phosphoesters), poly(alkyl-cyanoacrylates), poly(hyaluronic acids), and natural sugars such as chitosan, have been proposed as potential drug delivery implant materials [72]. Especially, biodegradable synthetic polymers, such as PLA, PGA, poly(ε-caprolactone) (PCL), polydioxanone, and their copolymers: (poly(lactic-*co*-glycolic acid) (PLGA), poly(lactic acid-*co*-ε-caprolactone) (PLACL) etc.), are commonly used in the PDDSs’ development due to their biocompatibility and tuneable degradation rates. These polymers degrade into nontoxic monomers that can be metabolically eliminated from the body, reducing the risk of long-term accumulation and adverse reactions. The degradation rate of these polymers can be controlled by modifying their average molecular weight (*M*_n_) and composition, allowing for the fine-tuning of drug release kinetics. For instance, a higher ratio of PLA to PCL in PLACL macromolecule results in a faster degradation rate and hence a quicker drug release. Furthermore, the surface properties of these polymers can be modified to enhance drug absorption and distribution. For example, the addition of hydrophilic or hydrophobic groups can improve the polymer’s interaction with the biological environment, affecting the drug’s release rate and targeting ability. Biodegradable polymers can also be formulated into various DDSs, such as nano- and microparticles, liposomes, hydrogels, and implants, each with their unique advantages and applications [73]. For instance, biodegradable polymer NPs can enhance drug stability, improve bioavailability, and allow for targeted drug delivery, while biodegradable polymer implants can provide long-term, localized drug release. Moreover, the use of biodegradable polymers in the PDDSs’ development has also opened up new possibilities for the delivery of sensitive drugs like proteins. These drugs can be encapsulated within the polymer matrix, protecting them from degradation and allowing for their controlled release. Due to the above, biodegradable polymers play a crucial role in PDDSs. Their biocompatibility, degradability, and versatility make them an ideal material for the design and development of effective DDSs. However, there are still challenges to overcome, such as the control of polymer degradation and drug release kinetics, the improvement of drug loading efficiency, and the enhancement of drug stability.

Over the years, numerous Xn polymeric formulations have been produced (Figure 2). The majority of them aim to improve Xn’s low water solubility and bioavailability, low stability, high photosensitivity, and short half-life. A few formulations have been related to its biotransformation in the gastrointestinal tract by hepatic enzymes, as well as the accumulation on the apical side of Caco-2 cells [42,65,66]. Consequently, Xn lacks the ability to produce an effective therapeutic response at the target site. As outcome of Xn’s limitations, it is necessary to develop a formulation that overcomes all of the limitations associated with Xn bioavailability and transport to the target site while remaining intact at the biotransformation sites. Innovative DDSs based on biodegradable polymers have opened up new avenues for delivering biological molecules to their target through eliminating key transportation barriers. Table 1 outlines the various PDDSs and naturally derived compounds that were previously utilised as platforms for Xn delivery.

Although Table 1 summarizes both in vitro and in vivo data, as well as additional studies conducted by the authors, I would like to highlight and investigate some additional interesting examples in which different biodegradable polymers and other natural carriers were used as Xn delivery platforms. Namely, Changwei Ma and coworkers proposed a novel carrier for improving the solubility and bioaccessibility of Xn using heat-stable, serpin-like protein Z (PZ) derived from barley malt [83]. By applying 200 μM PZ as an Xn carrier, increase in this compound’s solubility in water by 145-fold was observed compared to free Xn. Fluorescence results showed that Xn interacted with PZ through a static quenching process, with a binding constant (K_a_) of 5.70 × 10^4^ L/mol at 298 K. PZ, a serine protease inhibitor, demonstrated excellent digestive stability in vitro, inhibiting trypsin with a low IC_50_ of 3.90 ± 0.03 mg/L and caused only 26.54 ± 0.70% hydrolysis after gastrointestinal digestion. Furthermore, the retention ratio of Xn was 50.28 ± 1.80% after gastrointestinal digestion, which is eight-fold higher than Xn alone, indicating that PZ significantly improved Xn’s bioavailability.

Furthermore, a colon targeted liquisolid powder of Xn was developed using Central Composite Design (CCD) [84]. This formulation was developed using Transcutol P as the nonvolatile solvent, polysaccharide (guar gum and pectin) for colon targeting, and Syloid XDP (SXDP) as the coating agent. The optimised liquisolid formulation obtained from the Design Expert software demonstrated significant results by providing a sigmoid curve during in vitro evaluation, indicating that the optimised batch of Xn liquisolid is a colon targeted drug release formulation by restricting drug release for the first 5 h and causing burst release. As a result, the authors concluded that the GG-PCT Xn liquisolid formulation provided targeted drug release in the colonic region for the treatment of ulcerative colitis (UC). Additionally, the analytical techniques used to characterise the formulation revealed that Xn was completely soluble in the selected liquid, and that the liquid was completely adsorbing on the surface of SXDP and polysaccharides, resulting in the formation of an amorphous state. The stability investigation carried out under accelerated conditions revealed no significant difference between the fresh and aged formulations [84].

An interesting investigation on the use of synthetic biodegradable copolymer PLGA was conducted by Qiao and coworkers [85]. A co-solvent system of chloroform and dimethylformamide were used to develop an electropsun PLGA fibre membrane loaded with Xn. To improve its biological functionality as bone tissue engineering scaffolds, 5 wt.% hydroxyapatite-grafted poly(L-lactic acid) (HA-g-PLLA) was added to the spin solution. The goal of that study was to determine the effect of blending HA-g-PLLA on the properties of the medicated fibre membrane, such as the morphology, thermodynamics, wettability, drug release, mechanics, and cytotoxicity. It was discovered that incorporating HA-g-PLLA not only transformed the amorphous Xn/PLGA fibre membrane into a crystalline structure but also altered the membranous wettability. All types of medicated fibre membranes demonstrated sustained and controlled release profiles, and the ternary membrane blended with HA-g-PLLA had a slower drug release rate than the binary Xn/PLGA. The drug release rate in the ternary membrane increased with increasing Xn content. Tensile testing also revealed that at 10% of Xn, the comprehensive mechanics of the ternary outperformed the binary [85]. In a comparable manner, further research was conducted to obtain core/shell fibres containing resveratrol (RES) and Xn via coaxial electrospinning [86]. The authors discovered that RES/polyethylene oxide (PEO) could be successfully combined with Xn/PLGA to form core/shell fibres, but fibrous morphology deteriorated to some extent as drug content in the core or shell increased. The core/shell was amorphous, and changing the drug content had little effect on the wettability of the core/shell fibres. The release accelerated as the drug content increased in the fibres, and RES and Xn showed a gradient release, with RES releasing at a slower rate than Xn. These two drugs had a 350 h release window. The fibre mesh with 50/50 and 10/90 blend ratios of RES/PEO and Xn/PLGA had the best mechanical properties, with a tensile strength and elongation at break of 2.85 ± 0.10 MPa and 55.23 ± 2.53%, respectively [86]. As an alternative, the next study reported that the amorphous solid dispersions (ASDs) of Xn into biodegradable PCL containing up to 50 wt.% of the bioactive compound in amorphous formed as a consequence of the advantageous specific interactions established in this system [87]. Differential Scanning Calorimetry (DSC) was used in this work to explore the miscibility of PCL/Xn blends, which revealed the presence of favourable inter-association interactions. Still, the crystallinity results obtained using this technique were consistent with the results of X-ray diffraction (XRD) at the ambient temperature. Moreover, Fourier Transform Infrared Spectroscopy (FTIR) revealed strong C=O···O-H interactions between the active molecule’s hydroxyl groups and polymer chain’s carbonyl groups. It is also worth noting that the atomic force spectroscopy (AFM) analysis of the obtained materials demonstrated variations in a crystal morphology that depended on the composition.

Lastly, tensile tests have shown that blends in which Xn can be dispersed in amorphous form retain high toughness. The authors concluded that PCL was a convenient matrix for dispersing Xn in an amorphous form, allowing the development of completely amorphous bioactive materials suitable for non-stiff biomedical devices [87].

Aside from the most abundant Xn, hop cones contain a number of minimally structurally related prenylated chalcones, such as xanthogalenol, 4′-*O*-methylxanthohumol, 5′-prenylxanthohumol, and Xn B, C, D, E, and H [88]. One of the most promising is xanthohumol C (XnC). It has recently gained appeal due to its antiproliferative, cytotoxic, neuroprotective, and antioxidant properties [89]. Stevens and coworkers discovered XnC, also known as dehydrocycloxanthohumol, in the 1990s. XnC’s chemical structure differs from that of Xn by a ring closing of the phenyl side chain with the hydroxyl group at position 4′ [89]. It has previously been reported that numerous prenylated flavonoids have a significant antiproliferative and cytotoxic effect [90]. The authors observed that XnC exhibited a dose-dependent (0.1 to 100 μm) decrease in the growth of human breast cancer (MCF-7), colon cancer (HT-29), and ovarian cancer (A-2780) cells in vitro. After a 2-day treatment, XnC inhibited MCF-7 cell growth by 50% (IC_50_) at a concentration of 15.7 μm. Following a 4-day treatment, the IC_50_ for this compound was 6.87 μm. Furthermore, HT-29 cells were more resistant to this flavonoid than MCF-7 cells [90]. In addition, a recent study found that XnC promoted neuronal differentiation of neural stem cells and neurite outgrowth of cultured dorsal root ganglion neurons and protected neuronal rat pheochromocytoma cell line (PC12 cells) from cobalt chloride-induced, as well as cholinergic neurons of the nucleus basalis of Meynert from deafferentation-induced cell death [91].

Referring to the subject of innovative formulations, I would like to draw readers’ attention to the interesting work of Kirchinger and coworkers [92]. In this study, the researchers developed and verified a suitable XnC formulation for in vivo administration, namely the complex of XnC and 2-hydroxypropyl-β-cyclodextrin, which was thoroughly characterised using multiple analytical methods. It was discovered that XnC’s water solubility increases as cyclodextrin concentrations increase. Furthermore, in vitro bioactivity of XnC in free and complexed forms did not significantly differ, indicating that this active compound was released from 2-hydroxypropyl-β-cyclodextrin. In the end, a small-scale in vivo experiment in a rat model revealed that after the intraperitoneal (i.p.) administration of the complex, XnC was detected in serum, the brain, and the cerebrospinal fluid at 1 and 6 h. According to the authors, the formulation of XnC/2-hydroxypropyl-β-cyclodextrin was suitable for in vivo experiments and further pharmaceutical research aiming to determine its neuroregenerative potential in animal disease models [92].

## 4. Conclusions and Future Perspectives

Xanthohumol (Xn) is a remarkable nutraceutical with numerous pharmacological actions that has been extensively studied over the last few decades. Unfortunately, the US Food and Drug Administration (FDA) does not currently approve this active compound as a drug; however, it is available as a dietary supplement and an ingredient in medical foods. Despite numerous studies demonstrating Xn’s anti-infective, anti-obesity, anti-osteoporotic, anti-oxidative, chemopreventive, anti-inflammatory, and pro-apoptotic effects in cancer cell lines and tumour cells, its low water solubility and stability, high photosensitivity, and short half-life reduce bioavailability and gastrointestinal absorption. This has prompted researchers to conduct studies with various formulation strategies to improve its solubility and permeability. Another important factor limiting Xn application is that both Xn and its metabolites are primarily excreted in faeces within 24 h of administration. This review mainly focused on different carriers based on biodegradable polymers and other natural compounds used for Xn delivery and discussed the properties of the obtained materials along with in vitro and in vivo outcomes. Furthermore, wide biological properties of Xn were presented to highlight this molecule’s unique nature, including discussions on its safety, biotransformation, pharmacokinetics, and clinical applications. The conclusions drawn from this review clearly reflect the fact that only a few Xn nanoformulations based on biodegradable polymers have been obtained so far. This opens up new possibilities for scientists, as Xn’s pharmacological and antimicrobial properties can be applied in both the pharmaceutical and cosmetic industries.

The advances in nanotechnology have aided the expansion of a wide range of research areas. Finding the right kind of carriers, particularly those based on biodegradable polymers or naturally derived and nontoxic compounds, plays an essential role in the development of innovative and safe drug formulations. The efficacy of carriers with essential features (such as biocompatibility, efficiency of encapsulation, surface modification, biodegradability, etc.) is also being investigated. To achieve prolonged and controlled drug delivery release, various innovative formulations such as nanoparticles, nanomicelles, liposomes, solid lipid nanoparticles, and so on were investigated. The present review examines biodegradable polymers and other naturally derived compounds for Xn delivery and summarises their outcomes. As we can see from the presented data, the use of such platforms to deliver Xn is still in early stages of development, and the regulatory organisations have not yet granted full approval. However, developing innovative platforms based on natural and nontoxic compounds for delivering Xn can be challenging as well as expensive as we scale up for large-scale manufacturing.

Importantly, all of the presented studies have suggested that Xn is a noteworthy compound that could be used as an alternative strategy for the prevention and treatment of various diseases. However, most studies to date have only been conducted in in vitro and in vivo models. Nevertheless, these studies have demonstrated that this compound has a high potential. Consequently, these investigations have shown the importance of this compound against a variety of diseases, necessitating further preclinical and clinical research in the future to explore its tremendous therapeutic potential.

## Figures and Tables

**Figure 1 ijms-25-03398-f001:**
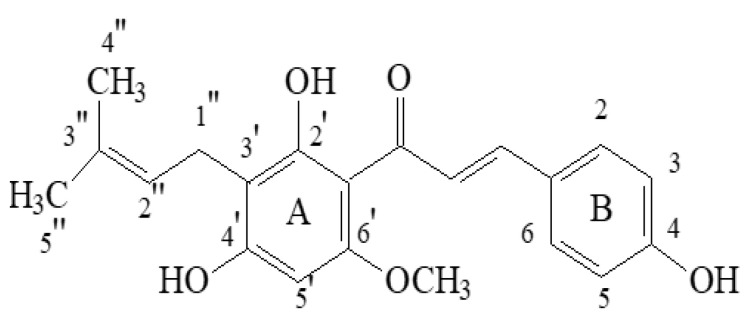
Chemical structure of xanthohumol (Xn).

**Figure 2 ijms-25-03398-f002:**
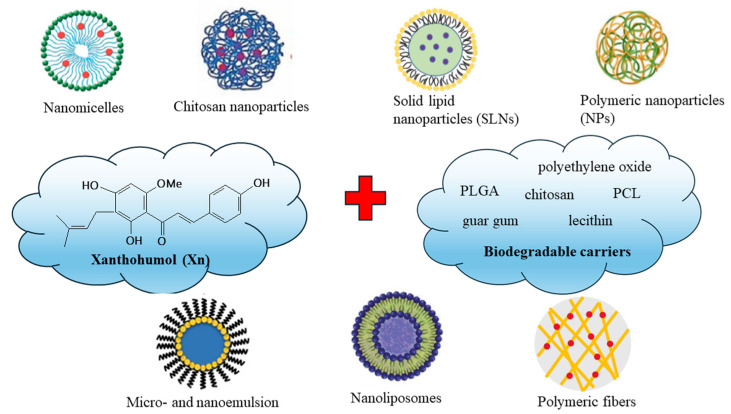
Different formulations for xanthohumol (Xn) delivery.

**Table 1 ijms-25-03398-t001:** Various drug delivery systems (DDSs) as the carriers for Xn delivery.

No	Xanthohumol (Xn) Formulation	Material Used	Particle Size [nm]	Entrapment Efficiency (*EE*) [%]	Zeta Potential (ζ) [mV]	Model Used	Outcomes	Ref
1.	Nanoparticles (NPs)	PLGA	191.0 ± 0.8	13.1 ± 0.06	−24.8 ± 0.2	Human corneal epithelial (HCE-T) cells, the mouse desiccating stress/scopolamine model and corneal epithelial cells.	Pure Xn prevented tert-butyl hydroperoxide-induced loss of cell viability in HCE-T cells in a dose-dependent manner and significantly increase in expression of the transcription factor nuclear factor erythroid 2-related factor 2 (Nrf2). Xn-loaded PLGA NPs were cytoprotective against oxidative stress in vitro, significantly reduced ocular surface damage and oxidative stress associated DNA damage in corneal epithelial cells in the mouse desiccating stress/scopolamine model for dry eye disease in vivo.	[74]
2.	Nanoparticles (NPs)	PLGA	312 ± 49	88.7 ± 4.3	−18.2 ± 1.4	B16F10, malignant cutaneous melanoma, and RAW264.7, macrophagic, mouse cell lines	Similar viability cytoxicity between pure Xn and Xn-loaded PLGA NPs at 48 h with the IC_50_ at 10 μM. Similar antimigration effects for pure Xn and Xn-loaded PLGA NPs. The M1 antitumour phenotype was stimulated on macrophages. The ultimate anti-melanoma effect emerges from an association between the viability, migration, and macrophagic phenotype modulation.	[75]
3.	Nanomicelles	Pluronic P123, F127 (polyethylene oxide (PEO) and polypropylene oxide (PPO) (PEO-PPO-PEO block copolymer)	30.4 ± 0.1–30.5 ± 0.2 (varied in Xn to Pluronic ratio)	93.5–100.0 (varied in Xn to Pluronic ratio)	−4.26–+4.26 (varied in Xn to Pluronic ratio)	Human colon cancer cell (HT-29 cells)	An enhanced in vitro cytotoxic activity was not achieved (it was associated with a lower polymer content in the tested formulations).	[76]
4.	Nanoliposomes	Lecithin	149.5–394.4	67.81	−19.9–−38.2	Antioxidant activities determined by DPPH method	The IC_50_ value for antioxidant activity was calculated to be 54.90 and 60.38 μg/mL for free Xn and nanoliposomal Xn.	[77]
5.	Solid self-nanoemulsifying drug delivery system (S-SNEDDS)	Liquid (L)-SNEDDS composed of Labrafac PG, Tween 80 and Transcutol P was adsorbed onto the surface of guar gum and pectin to form S-SNEDDS	118.96 ± 5.94	94.20 ± 4.71	−22.78 ± 1.13	Human colorectal adenocarcinoma (Caco-2) cell lines	43.90 ± 3.98% and 31.98 ± 3.10% cells were found viable at the end of 12 and 24 h. Increase in cytotoxicity of Xn-loaded S-SNEDDS was about 1.42-fold and 1.51-fold at the end of 12 h and 24 h as that of cytotoxicity of raw Xn.	[78]
6.	Solid lipid nanoparticles (SLNs)	Compritol E ATO (CE), precirol ATO5, Lipoid E 80SN (LE-80), lipoid S75, phospholipon 90H, pluronic F-68, tween 80, glyceryl monostearate (GMS), carnauba wax, palmitic and stearic acids	108.60 ± 3.21	80.20 ± 2.95	−12.7	Prostate cancer cell lines (PC-3); in vivo pharmacokinetic study; cell permeability studies	The cell inhibition percentage was noted in a dose- and time-dependent manner; a significant (*p* < 0.05) reduction in inhibitory activity was observed in the case of Xn-SLNs as that of naive Xn in the first 12 h; at 5 h, the release of Xn from Xn-SLNs was found to be 6.2 ± 0.98 nmol, whereas naive XH showed only 1.34 ± 0.087 nmol permeation (the permeation of Xn from Xn-SLNs was about 4.62-fold higher than naive Xn and 0.31-fold-lower drug excretion than naive Xn; the enhancement in the bioavailability of the Xn was confirmed from an increase in C_max_ (1.07-fold), AUC_0-t_ (4.70-fold), t_1/2_ (6.47-fold), and MRT (6.13-fold) after loading into SLNs; the relative bioavailability of Xn-SLNs and naive Xn was found to be 4791% and 20.80%.	[79,80]
7.	Microemulsion (ME) delivery system based on biosurfactant sophorolipids (SLs)	Tween 20/40/60/80, Span 20/80, isoamyl acetate (IA), SLs (lactonic), medium-chain triglyceride (MCT)	15.79–17.62	96.52 ± 0.23–97.51 ± 0.89	-	The in vitro digestion model	The antioxidant properties of Xn-SL-MEs were reflected in the IC_50_ values, with the decreases of 57.76% (DPPH), 83.94% (ABTS), and 34.34% (⋅OH); increase the solubility of Xn by about 4000 times, and its half-life during storage was extended to over 150 days; in vitro models revealed that the release profile of Xn followed non-Fickian diffusion, and the ME structure markedly strengthened its digestive stability and bioaccessibility.	[81]
8.	Nanochitosan (CNP) matrices encapsulated hop extracts	Chitosan (CS)	28.1–49.4 (varied in different synthesis conditions)	81.00–97.16 (varied in different molar weight of chitosan)	−17.6–47.3 (varied in different synthesis conditions)	*S. aureus*, *S. epidermis*, *P. aeruginosa*, *P. putida*, *C. albicans*, *C. lipolytica*	The decrease in MIC values (by factors from 1.25 to 6) was higher for the Gram-negative and for the *Candida* species; increased antimicrobial effect against a broad spectrum of species and remarkable synergistic interactions between chitosan and hops.	[82]

## Data Availability

Not applicable.

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
