# Peer review of "Xanthohumol—A Miracle Molecule with Biological Activities: A Review of Biodegradable Polymeric Carriers and Naturally Derived Compounds for Its Delivery"

_ijms, 2024, doi:10.3390/ijms25063398_

Round 1

Reviewer 1 Report

Comments and Suggestions for Authors

In the introduction, the author thoroughly describes both the chemical and biological aspects of xanthohumol and its derivatives. This is a well-written introduction to the topic. This news is then expanded upon in section 2. It takes up almost 6 pages. In my opinion, the main part should be section 3, relating to the topic of the manuscript, i.e. biodegradable polymers used as Xn carriers, which is 8 pages, almost half of which (4 pages are a table). The summary provides a concise summary of the most important information. 32 out of 89 references are from 2019-2023. Noteworthy is the current information relating to biodegradable polymers used as Xn carriers - 14 out of 19 references in section 4.

According to "iThenticate report" - many fragments of the article are almost copied fragments of the source publications, and this should not be the case. The author should write these passages again, in her own words.

Main remarks:

1) The title should be changed. As it stands, it suggests that the manuscript deals mainly with biodegradable Xn carriers. It lacks information regarding the discussed biological properties of Xn, which constitutes almost half of the manuscript's content.

2) Figure 1 - I am asking for better quality, in particular the alignment of the presented structure.

3) Lines 132-137: please write more simply. Focusing on the substituents attached to each rings. Moreover, the presented description is almost a copy of the description from section 5 of the article: https://iubmb.onlinelibrary.wiley.com/doi/10.1002/iub.2522

4) In many places in the article (e.g. lines 207, 209, 213, 214, 236, 238, 251, 306, 494, table 1) there is "ml" or "l" instead of the corresponding "mL" and "L". Please correct.

5) Line 240: "lately tested" - it cannot be said that ref 43 from 2004 is from recent years. Similarly, line 468 - "in recent years" - cannot be used to describe publications from 2004, 2004, 2009 (refs 39, 62, 63). Please rewrite in another form.

6) Line 209: "bitter acids (α- and β-acids)" - unnecessary repetition. Please leave "bitter acids" or "α- and β-acids"

7) Figure 2 - better quality please.

8) The "Abbreviations" section lacks an explanation of DPPH and ABTS. Moreover, there is a repetition in lines 669 and 670. Line 688 - NPs, please expand as "Nanoparticles" and not "Polymer Nanoparticles".

Comments on the Quality of English Language

Moderate editing of English language required

Author Response

Reviewer 1:

In the introduction, the author thoroughly describes both the chemical and biological aspects of xanthohumol and its derivatives. This is a well-written introduction to the topic. This news is then expanded upon in section 2. It takes up almost 6 pages. In my opinion, the main part should be section 3, relating to the topic of the manuscript, i.e. Biodegradable polymers used as Xn carriers, which is 8 pages, almost half of which (4 pages are a table). The summary provides a concise summary of the most important information. 32 out of 89 references are from 2019-2023. Noteworthy is the current information relating to biodegradable polymers used as Xn carriers - 14 out of 19 references in section 4. According to "iThenticate report" - many fragments of the article are almost copied fragments of the source publications, and this should not be the case. The author should write these passages again, in her own words.

Answer:

I would like to express my gratitude to the Reviewer for his/her suggestions and constructive criticism. They will undoubtedly help to improve the value of my work.

I would like to highlight that my article is a review of the literature on xanthohumol, its chemical and biological properties, and, most crucially, biodegradable polymeric carriers and naturally occurring substances for its delivery. I wrote the article in my own words and opinions, however, some of the most necessary conclusions were taken from the original source, but rephrased. In my opinion, we are unable to include several chemical and biological terminologies in the text differently from our predecessors because they are standard. I regret, but I disagree with the Reviewer's statement that numerous portions of my article were nearly copied from the source articles. I want to emphasise again that I have written my review article myself and the literature, fragments that I included in my  paper were rephrased.

Main remarks:

  1. The title should be changed. As it stands, it suggests that the manuscript deals mainly with biodegradable Xn carriers. It lacks information regarding the discussed biological properties of Xn, which constitutes almost half of the manuscript's content.

Answer:

I would like to thank the Reviewer for her/his constructive criticism of my work. The manuscript's title has been changed to "Xanthohumol - a miracle molecule with biological activities. A review of biodegradable polymeric carriers and naturally derived compounds for its delivery", according to the Reviewer's suggestion.

2. Figure 1 - I am asking for better quality, in particular the alignment of the presented structure.

Answer:

Figure 1 has been redrawn with a higher resolution in response to the Reviewer's feedback.

3. Lines 132-137: please write more simply. Focusing on the substituents attached to each rings. Moreover, the presented description is almost a copy of the description from section 5 of the article: https://iubmb.onlinelibrary.wiley.com/doi/10.1002/iub.2522

Answer:

The indicated manuscript fragment has been written in a more consistent manner and in my own words, as suggested by the Reviewer (lines 135-140).

4. In many places in the article (e.g. lines 207, 209, 213, 214, 236, 238, 251, 306, 494, table 1) there is "ml" or "l" instead of the corresponding "mL" and "L". Please correct.

Answer:

I have corrected the entire manuscript in accordance with the Reviewer's comment.

5. Line 240: "lately tested" - it cannot be said that ref 43 from 2004 is from recent years. Similarly, line 468 - "in recent years" - cannot be used to describe publications from 2004, 2004, 2009 (refs 39, 62, 63). Please rewrite in another form.

Answer:

Obviously, I agree with the Reviewer's criticism and apologise for using inappropriate language. I have rewritten the words indicated by the Reviewer to make them more accurate. Please see the lines 265-266 and 498.

6. Line 209: "bitter acids (α- and β-acids)" - unnecessary repetition. Please leave "bitter acids" or "α- and β-acids".

Answer:

The term "bitter acids" has been removed from the manuscript text, as requested by the Reviewer (see the line 232). Thank you.

7. Figure 2 - better quality please.

Answer:

The quality of Figure 2 has been improved in response to the Reviewer's criticism.

8. The "Abbreviations" section lacks an explanation of DPPH and ABTS. Moreover, there is a repetition in lines 669 and 670. Line 688 - NPs, please expand as "Nanoparticles" and not "Polymer Nanoparticles".

Answer:

I've added an explanation for the DPPH and ABTS in “Abbreviations”. Furthermore, I have deleted the repetition from line 698, and the abbreviation NPs expanded to "Nanoparticles". I would like to thank the Reviewer for providing helpful comments.

Reviewers and Editors

The language has been again reviewed and corrected by a native speaker.

The added or corrected text has been marked by yellow background (please see the corrected manuscript).

Reviewer 2 Report

Comments and Suggestions for Authors

The review article is very well-written and provides updates on research in the field. From my side, the article can be published with the inclusion of two minor revisions:

-Since the review article is focused on Xanthohumol, it would be interesting to dedicate a section to present the results of the molecule's physicochemical characterization. This could include experiments such as spectroscopy, calorimetry, and others.

-The visualization of Table 1 is not good. The author should adjust the table to fit the pages better.

Comments on the Quality of English Language

Minor editing of English language required

Author Response

Reviewer 2:

Comments and Suggestions for Authors

The review article is very well-written and provides updates on research in the field. From my side, the article can be published with the inclusion of two minor revisions:

1. Since the review article is focused on Xanthohumol, it would be interesting to dedicate a section to present the results of the molecule's physicochemical characterization. This could include experiments such as spectroscopy, calorimetry, and others.

Answer:

I would like to thank the Reviewer for this suggestion. I have added the physicochemical characterisation of xanthohumol per the Reviewer's recommendation. The characterisation covers its solubility in common solvents, elementary analyses, and spectroscopic and spectrophotometric data. In addition, I`ve included the data from a new research article on Xn's storage stability and degradation mechanism. Please refer to lines 128-158.

2. The visualization of Table 1 is not good. The author should adjust the table to fit the pages better.

Answer:

I have enhanced the visualisation of Table 1 by arranging it horizontally. I believe it is more readable now. I'd like to thank the Reviewer for his/her constructive criticism of my work.

Reviewers and Editors

The language has been again reviewed and corrected by a native speaker.

The added or corrected text has been marked by yellow background (please see the corrected manuscript).

Round 2

Reviewer 1 Report

Comments and Suggestions for Authors

Thank you very much for all the clarifications, explanations and corrections.

Final remarks:

1) Line 132: the melting temperature of 172'C is hard to consider as "high". I would suggest removing the word "high".

2) Xn can also be crystallized from dichloromethane - DOI: 10.1007/s00044-017-1887-9

3) Line 139: should be "trans" (italic)

4) Lines 144-155 are redundant. The work does not cover Xn structural data obtained using spectroscopic methods. Please delete.

In my opinion, after above-mentioned changes, the work will be suitable for publishing in IJMS MDPI.

Author Response

Reviewers` Comments to the Authors:

Reviewer 1:

Thank you very much for all the clarifications, explanations and corrections.

Final remarks:

  • Line 132: the melting temperature of 172'C is hard to consider as "high". I would suggest removing the word "high".

Answer:

Thank you for providing this comment. I've removed the word "high" per the Reviewer's request.

  • Xn can also be crystallized from dichloromethane - DOI: 10.1007/s00044-017-1887-9

Answer:

As I've explained to the Reviewer in my first letter, my article is a literature review on xanthohumol, its chemical and biological properties, and, most importantly, biodegradable polymeric carriers and naturally occurring substances for its delivery. In my manuscript, I've claimed that this molecule is insoluble in water and petroleum ether, but can be crystallised in 50% alcohol, 50% acetone, acetic acid, chloroform, benzene, and toluene - a total of six solvents.

I agree with the Reviewer that xanthohumol can be crystallised from dichloromethane, but this information is not crucial to my article. Furthermore, the literature reference indicated by the Reviewer  (DOI: 10.1007/s00044-017-1887-9, Anna K. Å»oÅ‚nierczyk et al. Medicinal Chemistry Research, 2017, 26, 1764) does not mention the crystallisation of xanthohumol from dichloromethane, but rather refers to the previous authors' work: AnioÅ‚ M, SzymaÅ„ska K, Å»oÅ‚nierczyk A. Tetrahedron, 2008, 64, 9544. ("The crude XN was purified by column chromatography and crystallisation, as previously described (AnioÅ‚ et al. 2008)" ).

In my point of view, there is no need to include an additional literature reference in my review paper.

  • Line 139: should be "trans" (italic)

Answer:

I have corrected the word "trans" in italics.

  • Lines 144-155 are redundant. The work does not cover Xn structural data obtained using spectroscopic methods. Please delete.

Answer:

One of the Reviewers suggested to include xanthohumol spectroscopic data into the manuscript text.

In response to his/her request, I've added the 1H, 13C NMR, and FTIR data for xanthohumol. These data were highlighted in yellow in the revised version of my work.
I apologise for disagreeing with the Reviewer. My manuscript covers xanthohumol structural data collected by spectroscopic techniques, as the 1H, 13C NMR, and FTIR are the spectroscopic techniques.

The added or corrected text has been marked by green background (please see the corrected manuscript).